# DGSM-SCAM-GAT AND MMT-ViT: MULTIMODAL AND GRAPH-BASED MALWARE DETECTION

## ABSTRACT

Malware detection encounters substantial challenges in real-time and multi-class tasks, as single-modality methods struggle to capture intricate behavioral patterns. To mitigate these limitations, we introduce two complementary models: DGSM-SCAM-GAT and MMT-ViT. The DGSM-SCAM-GAT model integrates dynamic gating, contextual aggregation mechanisms, and graph attention networks (GAT) to enhance temporal and structural modeling of API call sequences. Trained on a dynamic API call sequence dataset, it attains an accuracy of 99.31% and an F1-score of 99.64%, surpassing CNN-LSTM (accuracy: 98.92%). The MMT-ViT model employs multi-modal attention mechanisms and the pre-trained ViT architecture to effectively fuse features from assembly instruction sequences, binary grayscale images, and binary wavelet sequence features. Evaluated on a public dataset, it achieves 99.54% accuracy and 99.55% F1-score, outperforming Malcse (accuracy: 98.94%). Furthermore, ablation studies validate the critical contributions of individual modules, while comparative experiments underscore the superiority of our proposed models over state-of-the-art baselines. The detection frameworks developed in this study facilitate robust dynamic and static malware identification, with code available in the supplementary materials.

## 1 INTRODUCTION

The increasing complexity of network threats has enabled malware to evade traditional detection through dynamic loading and code obfuscation, posing severe risks to individual, enterprise, and national security Akhtar (2021). Dynamic analysis captures API call sequences for real-time monitoringSalehi et al. (2017), while static analysis extracts binary file features for offline classificationRaff et al. (2017). However, existing research predominantly relies on single-modal approaches. RNNs and CNNs face limitations in sequence modelingTobiyama et al. (2016), and although static methods like Transformers or CNNs excel in multi-class tasks, multimodal fusion studies remain insufficientBai & Thirumaran (2024).

This paper introduces a malware detection framework comprising DGSM-SCAM-GAT (Dynamic Gated Sequence and Contextual Aggregation Graph Attention Network) and MMT-ViT (Multi-Modal Transformer with Vision Integration). DGSM-SCAM-GAT integrates Graph Attention Networks, Dynamic Gated Sequence Module (DGSM)Chung et al. (2014), and Contextual Aggregation Module (SCAM), efficiently modeling dynamic API behaviors and achieving 99.31% accuracy and a 99.64% F1 score in binary classificationChung et al. (2014)Veličković et al. (2018). MMT-ViT leverages multimodal TransformersVaswani et al. (2023) and Vision TransformersDosovitskiy et al. (2021) to fuse instruction sequences, grayscale images, and wavelet sequencesChawla et al. (2021)Author1 et al. (2021), reaching 99.54% accuracy and a 99.55% F1 score on the big2015 datasetRonen et al. (2018). Ablation studies confirm the contribution of each modalityJiang & Stamp (2025), with dynamic and static models complementing each other to provide a comprehensive cybersecurity solution.

Contributions:

1) Designed DGSM-SCAM-GAT, optimizing dynamic API sequence analysis with an F1 score of 99.64%.

2) Developed MMT-ViT, fusing multimodal features to achieve an F1 score of 99.55%.

3) Implemented a complementary approach for dynamic real-time monitoring and static offline detection.

## 2 RELATED WORK

Malware detection relies on dynamic and static analysis. Dynamic analysis captures API call sequences in sandboxes; early machine learning methods like SVM require manual feature engineering, limiting complex dependency modelingSchultz et al. (2001). Deep learning approaches, such as RNNs and LSTM variants, enhance API sequence analysisPascanu et al. (2015). Jang et reported a Bi-LSTM accuracy of 91.41%, though long-sequence modeling is hindered by gradient vanishingJang et al. (2020). CNNs address this by extracting local patternsTobiyama et al. (2016), with Catak et al.'s 1D-CNN achieving 95% accuracy using n-gram featuresCatak et al. (2020). Recently, GNNs have gained attention for capturing topological relationshipsWu et al. (2021); Aditya et al found LSTM (97.3% accuracy) outperformed GRU (56.05%)Aditya et al. (2021), while Gao et al.'s GCN reached 97% accuracy but struggled with long-range dependenciesGao et al. (2021).

Static analysis examines binary features like instruction sequences and grayscale images. Santos et al. achieved 94% accuracy with n-gram + SVM but relied on manual featuresSantos et al. (2013). Ni et al.'s Transformer modeled instruction sequences, attaining a 0.94 F1 score on big2015. Nataraj et al.'s grayscale CNN reached 98% accuracy, though limited by semantic information capture, Multimodal fusion integrates diverse static featuresNataraj et al. (2011); Vasan et al.'s CNN achieved a 95% F1 score, and Yakuri et al.'s ViT fusion reached 96% F1 score, yet neither utilized wavelet sequences. MMT-ViT fuses three modalities using ViT transfer learningCaron et al. (2021)and modal attention mechanisms, achieving a 99.55% F1 score, with ablation studies validating modal synergyBaltrušaitis et al. (2017).

## 3 METHODOLOGY

This study proposes a complementary framework integrating single-modal dynamic analysis and multimodal static analysis. DGSM-SCAM-GAT based on GNN, optimizes binary classification of API call sequences, while MMT-ViT leverages ViT transfer learning and attention mechanisms for nine-class classification. The models are independently designed to enhance real-time detection and complex classification, forming a synergistic solution. The following subsections detail the dynamic model DGSM-SCAM-GAT and the static model MMT-ViT.

### 3.1 DYNAMIC DETECTION MODEL WITH DUAL-BRANCH GRAPH NEURAL NETWORK

DGSM-SCAM-GAT targets API call sequences, employing dual branches (DGSM and SCAM) with Graph Attention Networks to optimize binary classification. The model begins by embedding API node features, followed by Initial-GAT to capture semantic importance of adjacent APIs and dynamically compute edge weights. Subsequently, DGSM and SCAM extract topological and contextual node information. Pre-fusion via multi-head attention integrates these features, and a second Graph Attention Network refines the fused representations to enhance detection of complex malware call patterns. Finally, graph pooling generates graph-level representations for classification, achieving an F1 score of 99.64%, with the model architecture depicted in Figure 1.

### 3.1.1 DATA PREPROCESSING AND GRAPH REPRESENTATION

The dataset includes 20,000 samples, each a length-100 unique API call sequence with 309 APIs, for malicious vs. benign binary classification. APIs are mapped to indices 0-308, and sequences $S = \{s_1, s_2, ..., s_t\} (t = 100)$ are converted to directed graphs G=(V,E), where $V = \{v_1, v_2, ..., v_t\}$ represents API nodes and $E = \{v_i, v_{i+1}\} (i = 1, 2, ..., t - 1)$ denotes temporal edgesAngelo Schranko De Oliveira (2021). Batch size B is merged via edge index offsets.

### 3.1.2 FEATURE EXTRACTION, FUSION, AND CLASSIFICATION

The model employs embedding, Initial-GAT, dual branches, pre-fusion and Second-GAT for feature extraction and fusion, followed by graph pooling and classification.

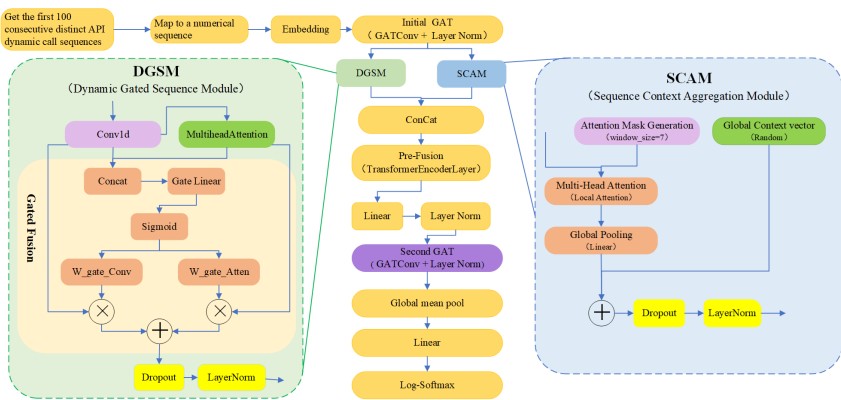

Figure 1: The DGSM-SCAM-GAT model.

1) Embedding Layer: The API indices $s_i \in [0, 308]$ are mapped to 256-dimensional vectors, $x_i = Embedding(s_i, W_e), where W_e \in \mathbb{R}^{309 \times 256}$ and $W_e$ is initialized using Xavier initialization.

2) Initial-GAT: For the high-dimensional vectors after embedding, an 8-head attention network is used to capture interactions between nodes, where the output of each head is given by Equation (1). The attention coefficients are calculated according to Equation (2), where $D_h = 256$. The outputs of the 8 heads are then concatenated and normalized, as shown in Equation (3).

$$h_i = \sigma(\Sigma_{j \in N_i} \alpha_{ij} W_{GAT1} x_j) \tag{1}$$

$$a_{ij} = \text{softmax}\left( \frac{(\mathbf{W}_{\text{GAT1}} \mathbf{x}_i)^{\top} (\mathbf{W}_{\text{GAT1}} \mathbf{x}_j)}{\sqrt{D_h/8}} \right), \quad \mathbf{W}_{\text{GAT1}} \in \mathbb{R}^{(D_h/8) \times 256} \tag{2}$$

$$\mathbf{H}_1 = \text{LayerNorm}\Big( \text{Concat}\big(\mathbf{h}_i^{(1)}, \mathbf{h}_i^{(2)}, \dots, \mathbf{h}_i^{(8)}\big) \mathbf{W}_0 \Big), \quad \mathbf{W}_0 \in \mathbb{R}^{256 \times 256} \tag{3}$$

3) Dual-Branch Module: Includes DGSM and SCAM

DGSM: Captures local temporal patterns using 1D convolution (kernel size 5, 256 channels, determined experimentally) as per Equation (4), models global dependencies with 8-head attention (Equation (5)), balances the feature weights dynamically via a gating mechanism (Equations (6) and (7)), and applies layer normalization (Equation (8)) to adapt to complex sequence patterns.

$$\mathbf{h}_{\text{conv}} = \text{Conv1D}\big(\mathbf{H}_1; \mathbf{W}_{\text{conv}}, k = 5\big), \qquad \mathbf{W}_{\text{conv}} \in \mathbb{R}^{256 \times 256 \times 5} \tag{4}$$

$$\mathbf{h}_{\text{attn}} = \text{MultiHeadAttention}(\mathbf{h}_{\text{conv}}, \mathbf{h}_{\text{conv}}, \mathbf{h}_{\text{conv}}), \quad \text{head} = 8, \, d = 256 \tag{5}$$

$$\text{Gate} = \sigma\Big( \text{Linear}\big([\,\mathbf{h}_{\text{conv}}; \mathbf{h}_{\text{attn}}\,]; \mathbf{W}_g\big) \Big), \qquad \mathbf{W}_g \in \mathbb{R}^{512 \times 256} \tag{6}$$

$$\mathbf{h}_{\text{dgsm}} = \text{Gate} \odot \mathbf{h}_{\text{conv}} + (1 - \text{Gate}) \odot \mathbf{h}_{\text{attn}} \tag{7}$$

$$\mathbf{h}_{\text{dgsm}} = \text{LayerNorm}\big(\text{Dropout}(\mathbf{h}_{\text{dgsm}})\big) \tag{8}$$

SCAM: Implements local attention with a sliding window, using a 7-window, 16-head attention mechanism (Equations (9) and (10)), followed by global pooling (Equation (11)). It then incorporates a learnable context vector c to enhance semantic context, forming global features $h_{\text{scam}}$ (Equation (12)), and applies normalization (Equation (13)), with output $h_{\text{scam}} \in \mathbb{R}^{B \times T \times 256}$.

$$\mathbf{h}_{\text{local}} = \text{LocalAttention}(\mathbf{H}_1, \mathbf{H}_1, \mathbf{H}_1, \text{mask} = M) \tag{9}$$

$$\text{MultiHeadAttention}(Q, K, V) = \text{Concat}\big(\text{head}_1, \ldots, \text{head}_H\big) W_0, \quad H = 16,$$

$$\text{head}_i = \text{Attention}(Q_i, K_i, V_i) = \text{Softmax}\left(\frac{Q_i K_i^\top}{\sqrt{d_k}} \odot (1 - M)\right) V_i, \tag{10}$$

$$M \in \mathbb{R}^{100 \times 100}, \quad \text{window\_size} = 7, \qquad \mathbf{W}_0 \in \mathbb{R}^{256 \times 256}.$$

$$\mathbf{h}_{\text{local}} = \mathbf{W}_{\text{pool}} \mathbf{h}_{\text{local}} + \mathbf{b}_{\text{pool}}, \quad \mathbf{W}_{\text{pool}} \in \mathbb{R}^{256 \times 256}, \ \mathbf{h}_{\text{local}} \in \mathbb{R}^{B \times T \times 256} \tag{11}$$

$$\mathbf{h}_{\text{scam}} = \mathbf{h}_{\text{local}} + \text{expand}(c, (B, T, D_h)), \quad c \sim \mathcal{N}(0, 1), \ \mathbf{h}_{\text{scam}} \in \mathbb{R}^{B \times T \times 256} \tag{12}$$

$$\mathbf{h}_{\text{scam}} = \text{LayerNorm}\big(\text{Dropout}(\mathbf{h}_{\text{scam}})\big) \tag{13}$$

4) Pre-Fusion Module: Concatenates the outputs of DGSM and SCAM, fuses them using a 16-head Transformer encoder (Equation (14)) for deep interaction, and projects the features to 256 dimensions via a projection layer to align with the subsequent Graph Attention Network (Equation (15)).

$$\mathbf{h}_{\text{fused}} = \text{TransformerEncoder}\big([\mathbf{h}_{\text{dgsm}}; \mathbf{h}_{\text{scam}}]; \text{heads} = 16, \ d = 512\big) \tag{14}$$

$$\mathbf{h}_{\text{fused}} = \text{Linear}(\mathbf{h}_{\text{fused}}; \mathbf{W}_{\text{fused}}), \quad \mathbf{W}_{\text{fused}} \in \mathbb{R}^{B \times T \times 256} \tag{15}$$

5) Second-GAT: Further captures temporal relationships in the fused features, enhancing representation, with the same formulation as Initial-GAT, yielding output $hi(2)$.

6) Graph Pooling and Classification: Employs global mean pooling to aggregate graph features (Equation (16)), followed by a linear layer to output binary classification probabilities, using log-softmax for stable prediction (Equation (17)).

$$\mathbf{h}_{\text{pool}} = \frac{1}{|V_b|} \sum_{i \in V_b} \mathbf{h}_i^{(2)}, \quad \mathbf{h}_{\text{pool}} \in \mathbb{R}^{B \times 256} \tag{16}$$

$$y = \text{logsoftmax}\big(\text{Linear}(\mathbf{h}_{\text{pool}}; \mathbf{W}_{\text{cls}})\big), \quad \mathbf{W}_{\text{cls}} \in \mathbb{R}^{2 \times 256} \tag{17}$$

## 3.2 STATIC DETECTION MODEL WITH MULTI-MODAL FEATURE LEARNING AND FUSION

The MMT-ViT model integrates assembly instruction sequences, binary wavelet sequences and binary grayscale images to achieve nine-class classification on the big2015 dataset. Feature extraction for assembly instruction and binary wavelet sequences relies on attention mechanisms, while grayscale images leverage transfer learning. The extracted features are stacked and deeply fused using a Fusion Module, enhanced with residual connections for stability, and classified via a fully connected layer. MMT-ViT achieves an F1 score of 99.55% on the big2015 dataset, with its architecture illustrated in Figure 2.

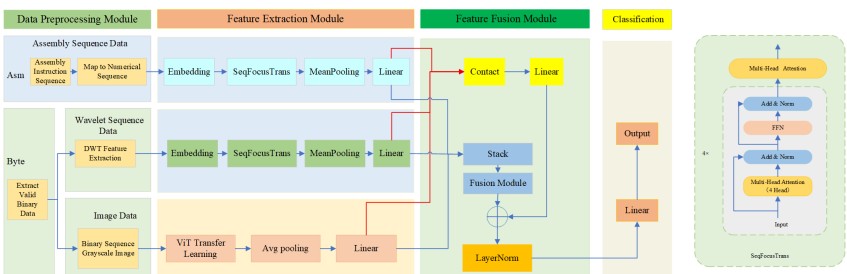

Figure 2: The DGSM-SCAM-GAT model.

### 3.2.1 ASSEMBLY SEQUENCE BRANCH

1) Data Preprocessing

The big2015 dataset contains 10,868 malware samples with assembly files. A vocabulary of 175 tokens filters valid instructions. Regular expressions remove address prefixes and bytecodes, deduplicate adjacent identical instructions, and set a maximum sequence length of 2048, with truncation or padding applied. Sequences are mapped to indices for input to the feature extraction module.

2) Feature Extraction

The assembly sequence branch processes instruction sequences (vocabulary size 175, $T_{\text{instr}} \leq 2048$), mapping input indices $t \in [0, 174]$ to 256-dimensional vectors via an embedding layer (Equation (18)). The SeqFocusTrans module, comprising 4-layer Transformer encoders (Equation (19)) and 4-head attention (Equation (20)), models long-range logical dependencies. Features are averaged via pooling (Equation (21)) and projected to 768 dimensions (Equation (22)), yielding $\mathbf{h}_{\text{instr}} \in \mathbb{R}^{B \times 768}$.

$$\mathbf{E}_{\text{instr}} = \text{Embedding}(x_{\text{instr}}), \qquad \mathbf{E}_{\text{instr}} \in \mathbb{R}^{B \times T_{\text{instr}} \times 256} \tag{18}$$

$$\mathbf{F}_{\text{instr}} = \text{TransformerEncoder}(\mathbf{E}_{\text{instr}};\ h = 4,\ l = 4,\ d_{\text{ff}} = 512), \quad \mathbf{F}_{\text{instr}} \in \mathbb{R}^{B \times T_{\text{inst}} \times 256} \tag{19}$$

where each Transformer layer (l=1,2,3,4):

$$\mathbf{H}_{\text{instr}}^{(l)} = \text{LayerNorm}\big(\text{MHA}(\mathbf{H}_{\text{instr}}^{(l-1)}) + \mathbf{H}_{\text{instr}}^{(l-1)}\big), \quad \mathbf{F}_{\text{instr}}^{(l)} = \text{LayerNorm}\big(\text{FFN}(\mathbf{H}_{\text{instr}}^{(l)}) + \mathbf{H}_{\text{instr}}^{(l)}\big),$$

$$\mathbf{H}_{\text{instr}}^{(0)} = \mathbf{E}_{\text{instr}}, \qquad \mathbf{F}_{\text{instr}} = \mathbf{F}_{\text{instr}}^{(4)}.$$

Multi-head attention (MHA):

$$Q_{\text{instr}},\ K_{\text{instr}},\ V_{\text{instr}} = \text{Linear}\big(\mathbf{H}_{\text{instr}}^{(l-1)};\ \mathbf{W}_Q, \mathbf{W}_K, \mathbf{W}_V\big),$$

$$\mathbf{head}_i = \text{softmax}\left(\frac{Q_{\text{instr},i} K_{\text{instr},i}^{\top}}{\sqrt{256/4}}\right) V_{\text{instr},i}, \qquad \mathbf{head}_i \in \mathbb{R}^{B \times T_{\text{instr}} \times 64},$$

$$\mathbf{A}_{\text{head}} = \text{Concat}\big(\mathbf{head}_1, \mathbf{head}_2, \mathbf{head}_3, \mathbf{head}_4\big)\mathbf{W}_0, \qquad \mathbf{W}_0 \in \mathbb{R}^{256 \times 256},$$

$$\text{MHA}\big(\mathbf{H}_{\text{instr}}^{(l-1)}\big) = \mathbf{A}_{\text{head}}, \qquad \mathbf{A}_{\text{head}} \in \mathbb{R}^{B \times T_{\text{instr}} \times D}.$$

Feed-forward network (FFN):

$$\text{FFN}(x) = \text{ReLU}(xW_1 + b_1)W_2 + b_2,$$

where $W_1 \in \mathbb{R}^{64 \times 512}, b_1 \in \mathbb{R}^{512}, W_2 \in \mathbb{R}^{512 \times 64}, b_2 \in \mathbb{R}^{64}$.

Layer normalization:$\text{LayerNorm}(x) = \frac{x-\mu}{\sqrt{\sigma^2+\varepsilon}} \odot \gamma + \beta$ ($\gamma,\beta$ are learnable parameters).

$$\mathbf{A}_{\text{instr}} = \text{MultiHeadAttention}(\mathbf{F}_{\text{instr}}, \mathbf{F}_{\text{instr}}, \mathbf{F}_{\text{instr}}; h = 4) \tag{20}$$

$$\mathbf{F}_{\text{instr}}^{\text{mean}} = \frac{1}{T_{\text{instr}}} \sum_{i=1}^{T_{\text{instr}}} \mathbf{A}_{\text{instr},i}, \qquad \mathbf{F}_{\text{instr}}^{\text{mean}} \in \mathbb{R}^{B \times 256} \tag{21}$$

$$\mathbf{h}_{\text{instr}} = \text{Linear}(\mathbf{F}_{\text{instr}}^{\text{mean}}; \mathbf{W}_{\text{instr-proj}}, \mathbf{b}_{\text{instr-proj}}), \quad \mathbf{W}_{\text{instr-proj}} \in \mathbb{R}^{256 \times 768} \tag{22}$$

### 3.2.2 WAVELET SEQUENCE BRANCH

Wavelet sequences extract time-frequency features from dataset byte files for the Wavelet branch of MMT-ViT. Hexadecimal byte sequences are read, invalid bytes (e.g., ??) are filtered, and sequences are converted to float arrays. A 3-level discrete wavelet transform with the db4 wavelet basis generates approximate coefficient cA3 and detail coefficients cD3, cD2, cD1, where cA3 represents low-frequency components and cD3, cD2, cD1 represent high-frequency components. Coefficients are normalized to a [2048, 4] sequence (2048 timesteps, 4-dimensional features: cA3, cD3, cD2, cD1) via zero-padding or truncation, capturing time-frequency patterns of malware. Two parallel processes enhance preprocessing efficiency.The feature extraction model and formulas for the wavelet sequence are identical to those used for the assembly sequence processing.

### 3.2.3 IMAGE BRANCH

Binary grayscale images convert malware byte streams into 2D representations to capture their spatial structural features. This study transforms one-dimensional data into 2D image data using a defined width table and adjusts the image resolution to [224, 224] to meet the input requirements of ViT transfer learning. To enhance model robustness, data augmentation is implemented through 10° random rotation, horizontal flipping (with a 50% probability of flipping along the vertical axis), and sequential combinations of these transformations to increase training data diversity. Detailed processing steps are provided in the appendix A.1.

Grayscale images are processed as 224×224 inputs, segmented into 16×16 patches, and encoded (Equation (23)). Transfer learning experiments with 2, 4 and 6 unfrozen layers culminate in a 4-layer pre-trained ViT extracting patch features (Equation (24)). Global features are derived via average pooling (Equation (35)) and projected to 768 dimensions (Equation (36)), yielding $\mathbf{h}_{\text{img}} \in \mathbb{R}^{B \times 768}$ from input $I \in \mathbb{R}^{B \times 1 \times 224 \times 224}$.

$$\mathbf{p}_{\text{img}} = \text{PatchEmbed}(I; \text{kernel} = 16, \text{stride} = 16), \quad I \in \mathbb{R}^{B \times 1 \times 224 \times 224}, \ \mathbf{p}_{\text{img}} \in \mathbb{R}^{B \times 196 \times 768} \tag{23}$$

$$\mathbf{h}_{\text{img}} = \text{ViTTransformer}(\mathbf{p}_{\text{img}}; \text{heads} = 12, d = 3072, \text{layers} = 12), \quad \mathbf{h}_{\text{img}} \in \mathbb{R}^{B \times 196 \times 768} \tag{24}$$

$$\mathbf{h}_{\text{img}} = \text{AvgPool}(\mathbf{h}_{\text{img}}; \dim = 1), \quad \mathbf{h}_{\text{img}} \in \mathbb{R}^{B \times 1 \times 768} \tag{25}$$

$$\mathbf{h}_{\text{img}} = \text{Linear}(\mathbf{h}_{\text{img}}; \mathbf{W}_{\text{img-proj}}), \quad \mathbf{W}_{\text{img-proj}} \in \mathbb{R}^{768 \times 768} \tag{26}$$

### 3.2.4 FEATURE FUSION AND CLASSIFICATION

The fusion module integrates features from the instruction sequence ($\mathbf{h}_{\text{instr}} \in \mathbb{R}^{B \times 768}$),wavelet sequence($\mathbf{h}_{\text{wav}} \in \mathbb{R}^{B \times 768}$),and grayscale image($\mathbf{h}_{\text{img}} \in \mathbb{R}^{B \times 768}$)branches (see Appendix A.2). Features are stacked along the sequence dimension (Equation(27)), modeled with an 8-head attention

mechanism for cross-modal interaction (Equation (28)), and further fused via a 6-layer Transformer encoder with average pooling (Equation (29)). Residual connections and layer normalization enhance stability by combining pooled and concatenated features (Equation(30) and (31)). A fully connected layer outputs nine-class probabilities (Equation (32)).

$$\mathbf{M}_{\text{stack}} = \text{Stack}\big([\,\mathbf{h}_{\text{instr}}, \mathbf{h}_{\text{wav}}, \mathbf{h}_{\text{img}}\,]; \dim = 1\big), \quad \mathbf{M}_{\text{stack}} \in \mathbb{R}^{B \times 3 \times 768} \tag{27}$$

$$\mathbf{A}_{\text{modality}} = \text{MultiHeadAttention}(\mathbf{M}_{\text{stack}}, \mathbf{M}_{\text{stack}}, \mathbf{M}_{\text{stack}}) \tag{28}$$

$$\mathbf{F}_{\text{fused}} = \text{TransformerEncoder}\big(\mathbf{A}_{\text{modality}}; \text{ heads} = 8, \ d = 2048, \text{ layers} = 6\big), \quad \mathbf{F}_{\text{fused}} \in \mathbb{R}^{B \times 3 \times 768} \tag{}$$

$$\mathbf{F}_{\text{fused}}^{\text{mean}} = \frac{1}{3} \sum_{i=1}^{3} \mathbf{F}_{\text{fused},i}, \qquad \mathbf{F}_{\text{fused}}^{\text{mean}} \in \mathbb{R}^{B \times 768} \tag{29}$$

$$\mathbf{F}_{\text{residual}} = \text{Linear}\big(\text{Concat}([\,\mathbf{h}_{\text{instr}}, \mathbf{h}_{\text{wav}}, \mathbf{h}_{\text{img}}\,]; \dim = 1); \mathbf{W}_{\text{res}}\big), \quad \mathbf{W}_{\text{res}} \in \mathbb{R}^{2304 \times 768} \tag{30}$$

$$\mathbf{F}_{\text{final}} = \text{LayerNorm}\big(\mathbf{F}_{\text{fused}}^{\text{mean}} + \mathbf{F}_{\text{residual}}\big) \tag{31}$$

$$\mathbf{y} = \text{Softmax}\big(\text{Linear}(\mathbf{F}_{\text{final}}; \mathbf{W}_{\text{cls}})\big), \quad \mathbf{W}_{\text{cls}} \in \mathbb{R}^{768 \times 9} \tag{32}$$

## 4 EXPERIMENTAL SETUP AND RESULTS

### 4.1 DGSM-SCAM-GAT EXPERIMENTAL SETUP AND RESULTS

#### 4.1.1 DGSM-SCAM-GAT DATASET AND MODEL PARAMETERS

The dataset used in this study is sourced from Kaggle, comprising 20,000 samples, each with a sequence length of 100 and containing 309 unique APIs. The dataset is split into 80% training and 20% testing sets in the laboratory environment.

The parameter settings for the DGSM-SCAM-GAT model are detailed in Appendix A.3.

#### 4.1.2 DGSM-SCAM-GAT EXPERIMENTAL RESULTS

1) Performance Evaluation

   Through model tuning and multiple ablation studies, the proposed DGSM-SCAM-GAT model achieved the following performance on the test set: Accuracy: 99.31%, Precision: 99.31%, Recall: 99.31%, F1-Score: 99.64%, ROC: 0.99. These results demonstrate excellent overall performance, with the high F1-Score indicating robustness on imbalanced datasets. The confusion matrix and ROC curve are shown in Appendix A.4.

2) Ablation Study and Comparative Experiments

   Ablation experiments were conducted by individually removing the DGSM module, SCAM module, GAT module, and pre-fusion module. The results indicate that all ablation configurations exhibit an accuracy degradation ranging from 0.09% to 0.21%, confirming the necessity of each module for achieving optimal performance,detailed results are presented in Table 1.

   Comparative experiments demonstrate that the proposed DGSM-SCAM-GAT model outperforms methods such as CNN-LSTM and DenseNet in malware detection, with improvements in accuracy of 0.39%–1.82%, F1-score of 0.19%–1.69%, and ROC of 0.01–0.13, results are summarized in Table 2.

The DGSM-SCAM-GAT model's generalization capability validation, computational complexity analysis, and innovation points are provided in the appendix A.4.

Table 1: Ablation Study Results Comparison

| Ablation Scheme | Accuracy (%) | Precision (%) | Recall (%) | F1 Score (%) | ROC |
|---|---|---|---|---|---|
| DGSM-SCAM-GAT | 99.31 | 99.31 | 99.31 | 99.64 | 0.99 |
| No DGSM | 99.19 | 99.18 | 99.19 | 99.57 | 0.97 |
| No SCAM | 99.10 | 99.09 | 99.09 | 99.52 | 0.96 |
| No GAT | 99.22 | 99.20 | 99.22 | 99.58 | 0.96 |
| No Pre-Fusion | 99.16 | 99.15 | 99.16 | 99.56 | 0.98 |

Table 2: Comparative Experimental Results

| Experimental Models | Accuracy (%) | Precision (%) | Recall (%) | F1 Score (%) | ROC |
|---|---|---|---|---|---|
| CNN_LSTM | 98.92 | 99.22 | 99.67 | 99.45 | 0.98 |
| DenseNet | 98.26 | 98.39 | 99.85 | 99.11 | 0.89 |
| ALPE | 98.40 | 98.45 | 99.92 | 99.18 | 0.92 |
| RBE | 98.31 | 98.38 | 99.90 | 99.14 | 0.86 |
| CNE | 98.07 | 97.98 | 98.07 | 97.95 | 0.98 |
| CNN+GAT | 97.49 | 97.31 | 97.49 | 98.68 | 0.96 |
| DGSM-SCAM-GAT | 99.31 | 99.31 | 99.31 | 99.64 | 0.99 |

## 4.2 MMT-ViT Experimental Setup and Results

### 4.2.1 MMT-ViT Dataset and Model Parameters

The dataset includes 9 malware families, with a total of 10,868 samples, comprising .bytes and .asm files. This study leverages this dataset to evaluate the classification performance of a multimodal model. Dataset details are provided in Table 5.

The parameter settings and Training Configuration Details for the MMT-ViT model are detailed in Appendix A.5.

### 4.2.2 MMT-ViT Experimental Results

1) Performance Evaluation

   Through model tuning and comparative experiments, the MMT-ViT model achieved the following performance on the test set:Accuracy: 99.54%,Precision: 99.55%,Recall: 99.54%,F1-Score: 99.55%. These results indicate excellent overall performance, with the high F1-Score demonstrating robustness on imbalanced datasets. The confusion matrix is presented in the appendix A.6.

2) Ablation Study and Comparative Experiments

   Ablation experiments were conducted by removing the wavelet sequences, instruction sequences, grayscale images, retaining only a single modality, or removing residual connections. The results indicate that all ablation configurations exhibit an accuracy degradation ranging from 0.09% to 22.95%, confirming the necessity of each module for achieving optimal performance, detailed results are presented in Table3.

   Comparative experimental results demonstrate that the proposed MMT-ViT model outperforms methods such as SSTL and MalCVs, with improvements in accuracy of 0.6%–4.54% and F1-score of 0.88%–6.92%, results are summarized in Table 4.

The MMT-Vit model's generalization capability validation, computational complexity analysis, and innovation points are provided in the appendix A.6.

Table 3: Ablation Study Results Comparison

| Ablation Experiment Schemes | Accuracy (%) | Precision (%) | Recall (%) | F1 Score (%) |
|---|---|---|---|---|
| MMT-ViT | 99.54 | 99.55 | 99.54 | 99.55 |
| Wavelet-Graph | 98.71 | 98.71 | 98.71 | 97.87 |
| Wavelet-Instr | 98.76 | 98.79 | 98.76 | 98.59 |
| Graph-Instr | 99.36 | 99.36 | 99.36 | 97.78 |
| Only Wavelet | 76.59 | 81.81 | 76.59 | 67.95 |
| Only Graph | 99.13 | 99.12 | 99.13 | 96.89 |
| Only Instr | 98.80 | 98.82 | 98.80 | 98.77 |
| No Modality | 99.31 | 99.32 | 99.31 | 87.61 |
| Attention-No Fusion | 99.45 | 99.45 | 99.45 | 98.62 |
| Transformer-No Residual | 99.31 | 99.32 | 99.31 | 99.22 |

Table 4: Comparative Experimental Results

| Author | Year | Model | Accuracy (%) | Precision (%) | Recall (%) | F1 Score (%) |
|---|---|---|---|---|---|---|
| Gao et al. | 2020 | SSTLe | 96.90 | 96.92 | 96.90 | 96.81 |
| Xiao et al. | 2021 | Malcse | 98.94 | 98.26 | 97.72 | 97.91 |
| Sudhakar et al. | 2021 | MCFT | 98.64 | 98.56 | 96.00 | 97.22 |
| Ravi et al. | 2023 | CNN-LSTM | 95.00 | 96.00 | 93.00 | 94.00 |
| Zou et al. | 2023 | FACILE | 97.20 | 93.99 | 92.14 | 92.63 |
| Xu et al. | 2024 | MDCAMe | 98.76 | 98.71 | 98.62 | 98.67 |
| **Our** | 2025 | MMT-ViT | 99.54 | 99.55 | 99.54 | 99.55 |

## 5 CONCLUSION

This study proposes two malware detection models: DGSM-SCAM-GAT and MMT-ViT. The DGSM-SCAM-GAT model is trained on a dynamic API call sequence dataset, constructing a graph attention network to leverage dynamic gating and contextual aggregation mechanisms for enhanced temporal and graph structural modeling of API sequences. The MMT-ViT model is trained on the BIG 2015 dataset, utilizing the base variant of the pre-trained ViT model along with multi-head attention modules to effectively fuse features from assembly instructions, wavelet coefficients, and grayscale images, thereby improving the model's feature extraction capabilities. Experimental results demonstrate that DGSM-SCAM-GAT and MMT-ViT achieve F1 scores of 99.65% and 99.55%, respectively, on their corresponding datasets. Ablation studies further validate the importance of each modal feature, while fine-tuning evaluations on new datasets also yield promising classification performance. Future work will explore lightweight model variants and enhancements to cross-dataset generalization capabilities.

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

# A  APPENDIX

## A.1  MALWARE GRAYSCALE IMAGE PROCESSING

Binary grayscale images transform malware byte streams into 2D representations to capture their spatial structural features. The process begins by reading binary files, parsing hexadecimal byte sequences row-wise, filtering invalid bytes (e.g., ??), and converting valid bytes to integers (0-255) to form a 1D byte array. To convert this array into a 2D image, image width and height are dynamically determined. This study defines a width table [32, 64, 128, 256, 384, 512, 640, 768, 896, 1024], corresponding to file size ranges [0-10KB, 10-20KB, 20-40KB, 40-80KB, 80-160KB, 160-320KB, 320-480KB, 480-800KB, 800-1000KB, ¿1000KB]. An appropriate width is selected based on the sample file size, with height calculated as height=len(bytes)/widthheight = len(bytes) / widthheight = len(bytes) / width, and the byte array is padded into a [height, width] matrix, with zeros filling deficiencies. This dynamic adjustment ensures structural consistency across files of varying sizes while preserving spatial byte distribution. Post-conversion, malware of the same type often exhibits similar texture patterns, as shown in Figure 3. During training, the 2D matrix is converted to an 8-bit unsigned integer array and resized to [224, 224] to align with ViT transfer learning input requirements. To enhance model robustness, data augmentation is applied, including 10° random rotation, horizontal flipping (with 50% probability along the vertical axis), and sequential combinations of these transformations to increase training data diversity, mitigating overfitting and improving robustness.

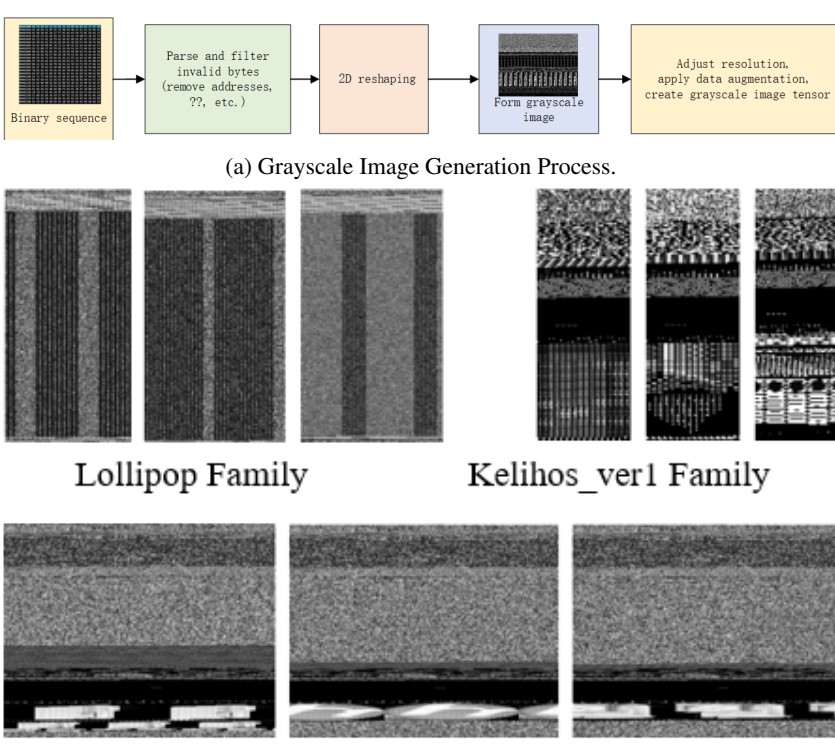

(a) Grayscale Image Generation Process.

(b) Grayscale Image Samples.

Figure 3: Binary Grayscale Image Samples of Different Malware Families

## A.2  MMT-ViT MODEL FEATURE FUSION

The architecture of the feature fusion in the MMT-ViT model is illustrated in Figure 4. In the feature fusion module, this component integrates features from the instruction sequence, wavelet sequence,

Table 5: DGSM-SCAM-GAT model Parameters

| Parameter | Value | Description |
|---|---|---|
| Embedding Dimension | 256 | Dimension of the embedded vector for API call indices |
| Batch Size | 16 | Number of samples per training batch |
| Hidden Layer Dimension | 256 | Dimension of hidden layers in each module |
| Number of Attention Heads | 8 | Number of heads in the multi-head attention module |
| Dropout | 0.3 | Prevents overfitting |
| Global Feature Count | 309 | Total number of distinct APIs |

and grayscale image branches. Initially, the features from the three branches are stacked along the sequence dimension. Subsequently, an 8-head attention mechanism models inter-modal information to enable cross-modal interaction, followed by a 6-layer fusion Transformer encoder for further feature integration and average pooling. The pooled features are then combined with concatenated features through residual connections and layer normalization to enhance stability. Finally, a fully connected layer outputs the nine-class classification probabilities.

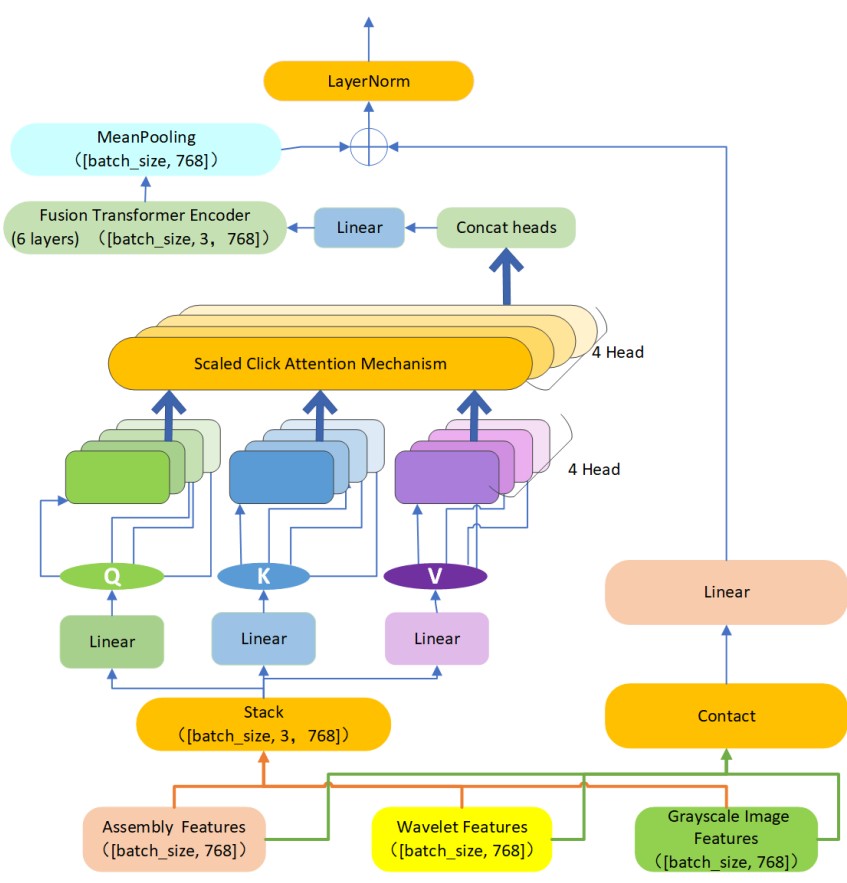

Figure 4: Multimodal Fusion Network Structure.

## A.3 THE PARAMETER SETTINGS FOR THE DGSM-SCAM-GAT MODEL

The parameter settings for the DGSM-SCAM-GAT model are detailed in table 5.

## A.4 ANALYSIS OF THE DGSM-SCAM-GAT MODEL

### A.4.1 GENERALIZATION VALIDATION

The Confusion matrix and ROC curve for the DGSM-SCAM-GAT model are shown in figure 5, the Loss and Accuracy for the DGSM-SCAM-GAT model are shown in figure 6. To evaluate the generalization ability, the model trained on the dynamic API call sequence dataset was tested on the mal-api-2019 dataset from Kaggle. The initial results were:Accuracy: 77.87%,Precision: 89.36%,Recall: 77.87%,F1-Score: 84.78%,ROC: 0.84.The Validation confusion matrix and ROC curve are shown in figure 7.

The performance degradation was primarily due to: Sequence Length Distribution,The mal-api-2019 dataset has an average API call sequence length of 150 (standard deviation 50), compared to 100 (standard deviation 20) in the training dataset. Feature Space Discrepancy, Approximately 30% of mal-api-2019 samples contain obfuscated code, which the DGSM module was not sufficiently exposed to during training.

To address these issues, transfer learning was applied by fine-tuning the model on the mal-api-2019 dataset for 10 epochs. The fine-tuned performance improved significantly:Accuracy:98.58%, Precision:98.58%, Recall:98.58%, F1-Score:99.15%, ROC:0.99. The fine-tuned confusion matrix and ROC curve are shown in figure 8.

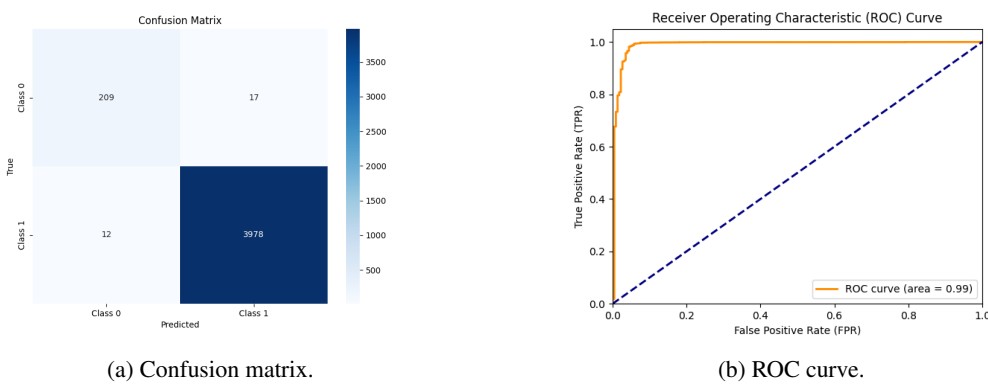

(a) Confusion matrix.                    (b) ROC curve.

Figure 5: Confusion matrix and ROC curve for the DGSM-SCAM-GAT model.

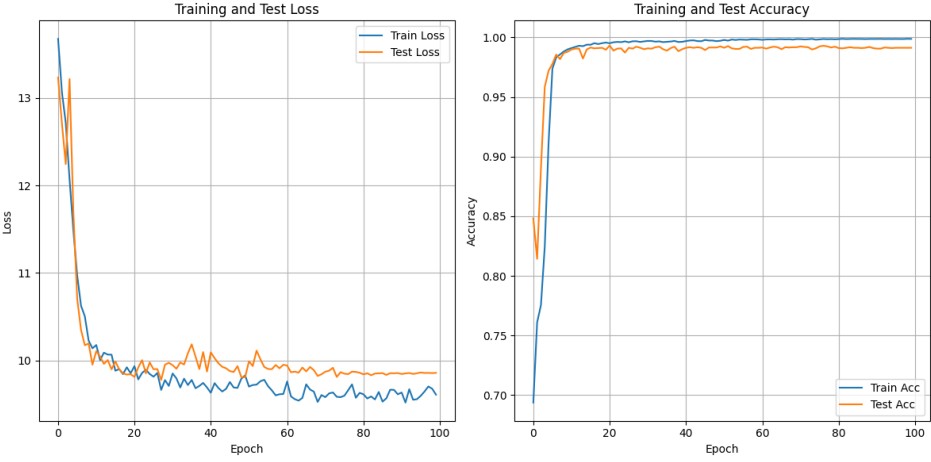

Figure 6: Loss and Accuracy for the DGSM-SCAM-GAT model.

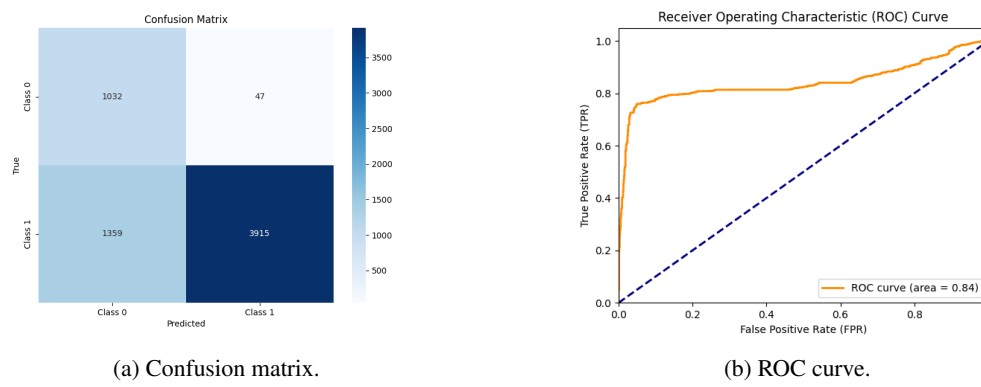

(a) Confusion matrix.                    (b) ROC curve.

Figure 7: Confusion Matrix and ROC Curve of the Validation Results.

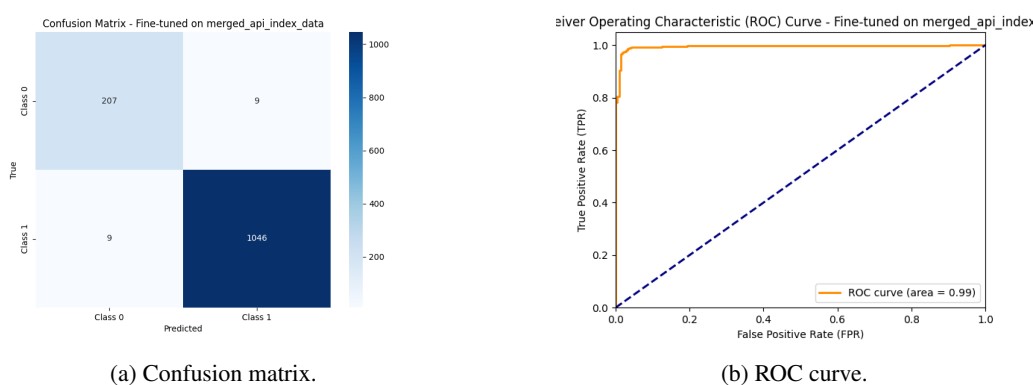

(a) Confusion matrix.                    (b) ROC curve.

Figure 8: Confusion Matrix and ROC Curve of the fine-tuned Results.

### A.4.2 COMPUTATIONAL COMPLEXITY ANALYSIS

To evaluate the computational cost of the DGSM-SCAM-GAT model, all experiments were conducted on an NVIDIA RTX 4070 GPU. The model integrates Graph Attention Networks (GAT), Dynamic Gated Sequence Module (DGSM), Sequence Context Aggregation Module (SCAM), and Transformer encoder layers, processing API call sequences of length 100 with a batch size of 16 and hidden layer dimension of 256. Training for 100 epochs resulted in a total computational cost of approximately $5.125 \times 10^{12}$ FLOPs, with a training time of approximately 28.49 minutes and a total parameter count of 3.49M, making it suitable for small to medium-scale datasets.

### A.4.3 DGSM-SCAM-GAT INNOVATIONS

The model employs a Graph Attention Network to capture the semantic importance of adjacent APIs, dynamically computing edge weights. It introduces dual-branch modules (DGSM and SCAM) for feature extraction and fusion, followed by a Transformer encoder and linear layer to enhance the model's ability to capture diverse patterns in API call sequences. During training, a weighted loss function addresses class imbalance, label smoothing improves generalization, and a warm-up with cosine decay scheduler optimizes learning, collectively enhancing model robustness and training stability.

### A.5 THE PARAMETER SETTINGS FOR THE MMT-VIT MODEL

Through parameter tuning, the final model parameter settings are detailed in Table 6. The training process employs the AdamW optimizer (initial learning rate of $1 \times 10^{-4}$, weight decay of $1 \times 10^{-5}$),

Table 6: Parameter Settings

| Module | Parameter Name | Parameter Value |
|---|---|---|
| Global | Number of Classes | 9 |
| Global | Hidden Dimension | 768 |
| Global | Attention Heads | 8 |
| Global | Fusion Layers | 6 |
| Instruction Sequence Encoding | Vocabulary Size | 151 |
| Instruction Sequence Encoding | Embedding Dimension | 256 |
| Instruction Sequence Encoding | Transformer Layers | 4 |
| Instruction Sequence Encoding | Transformer Heads | 4 |
| Instruction Sequence Encoding | Feed-Forward Dimension | 512 |
| Instruction Sequence Encoding | Dropout | 0.3 |
| Image Feature Extraction | ViT Model | vit-base-patch16-224 |
| Image Feature Extraction | Output Dimension | 768 |
| Image Feature Extraction | Freeze/Unfreeze | first 8 frozen, last 4 unfrozen |
| Feature Fusion | Modal Attention Heads | 8 |
| Feature Fusion | Fusion Transformer Layers | 6 |
| Feature Fusion | Feed-Forward Dimension | 2048 |
| Feature Fusion | Dropout | 0.3 |
| Classification | Fully-Connected Dimension | $768 \rightarrow 9$ |

combined with the OneCycleLR scheduler (maximum learning rate of $1 \times 10^{-3}$). Efficiency and performance are enhanced through gradient accumulation and mixed-precision training. The loss function uses cross-entropy loss with class weights, defined as $w_i = 1/\max(\text{Counter}(y_{\text{train}})[i], 1)$.

### A.6 ANALYSIS OF THE MMT-ViT MODEL

#### A.6.1 GENERALIZATION VALIDATION

The Confusion matrix for the Multimodal Fusion Network Structure is in Figure 9.The model, pre-trained on the big2015 dataset, was fine-tuned and validated on the new datasets Malimg (25 classes), Malevis, and Malimg (31 classes). Fine-tuning validation results on the Malimg dataset (25 classes) are as follows: accuracy of 98.69%, precision of 98.70%, recall of 98.69%, and F1 score of 96.13%, with the confusion matrix presented in Figure 10. For the Malevis + Malimg dataset (31 classes), the fine-tuning validation results are: accuracy of 93.06%, precision of 93.00%, recall of 93.06%, and F1 score of 93.35%, with the confusion matrix shown in Figure 11. Fine-tuning validation on these new datasets demonstrates the model's strong generalization capability.

#### A.6.2 COMPUTATIONAL COMPLEXITY ANALYSIS

To assess the computational cost of the MMT-ViT model, all experiments were conducted on an NVIDIA RTX 4070 GPU. The model integrates three modalities based on Transformer and ViT architectures, processing sequences of length 2048 and 224×224 pixel images. With a batch size of 8, hidden dimension of 768, and 20,000 samples trained for 100 epochs, the total computational cost is approximately $4.95 \times 10^{15}$ FLOPs, with a training time of 15.56 hours and a total parameter count of 83.62M, suitable for high-performance computing environments.

#### A.6.3 MMT-ViT INNOVATIONS

The model introduces a tri-modal feature fusion framework, leveraging an 8-head multimodal attention mechanism to dynamically capture inter-modal interactions. Compared to traditional single-modal or simple weighted fusion methods, this mechanism adaptively learns the contribution weights of each modality, effectively exploiting the semantic information of instruction sequences, temporal features of wavelet sequences, and visual patterns of grayscale images to improve clas-

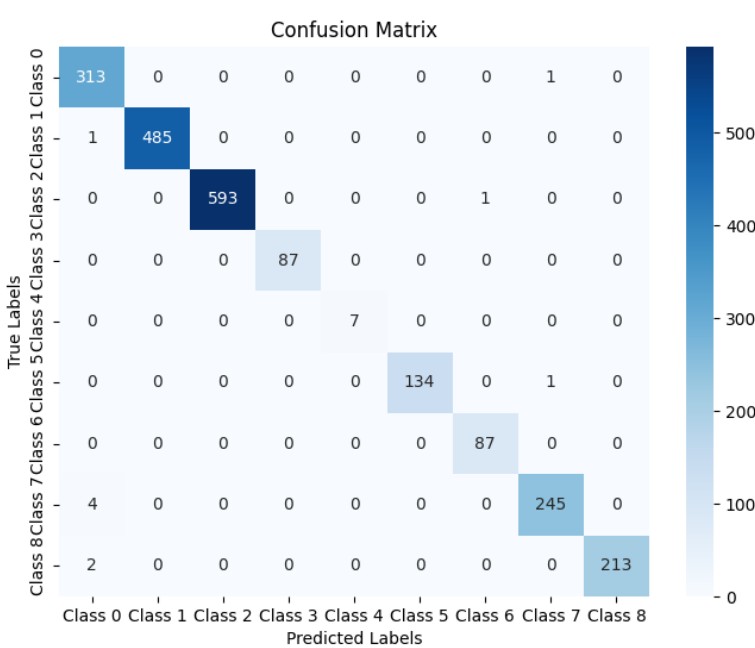

Figure 9: Confusion matrix for the Multimodal Fusion Network Structure.

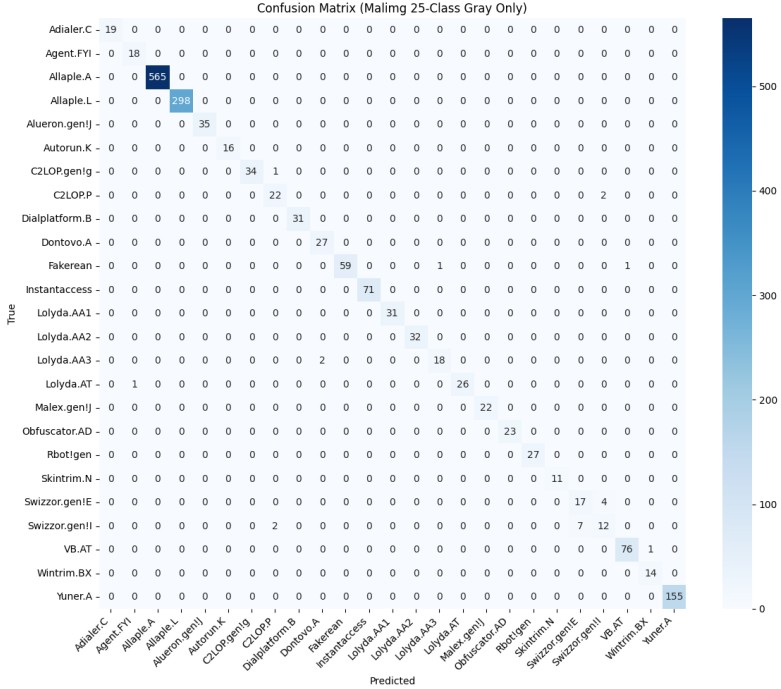

Figure 10: Confusion Matrix (Malimg, 25 classes).

sification accuracy.The deep Transformer fusion architecture, consisting of 6 layers with 8-head attention, captures complex nonlinear dependencies between modalities, enhancing feature representation robustness. Residual connections and layer normalization mitigate gradient vanishing

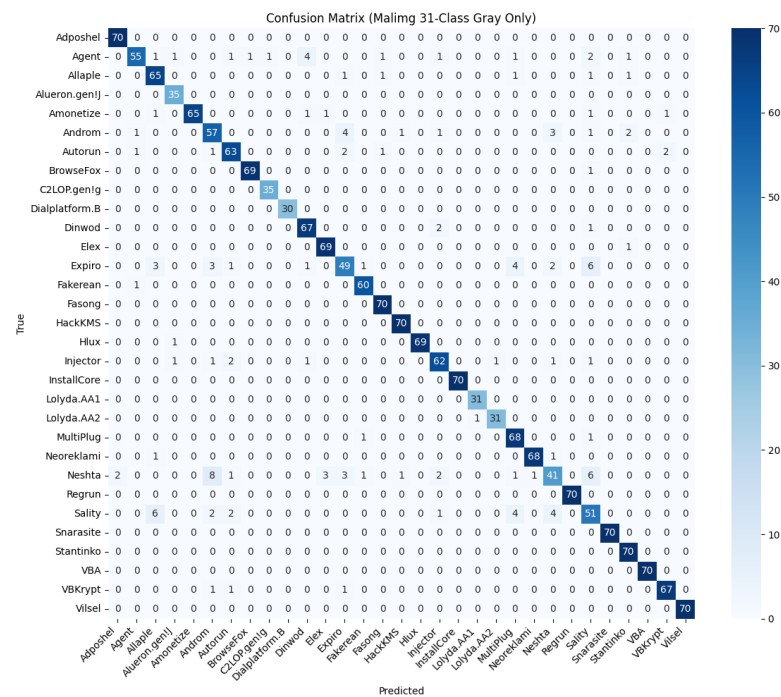

Figure 11: Confusion Matrix (Malevis + Malimg, 31 classes).

issues in deep networks, improving training stability.To address class imbalance in malware classification, a class-frequency-based weighted cross-entropy loss is adopted, assigning higher weights to rare classes to improve their classification performance. Combined with the OneCycle learning rate scheduler (maximum learning rate 0.001, 100 epochs), the training process is optimized for multi-class tasks, ensuring robust generalization.

## A.7 TOOLS AND RESOURCES

This research benefited from the use of the AI tool, which assisted in data search, paper polishing, and translation tasks. With AI's help, the overall quality of the paper was enhanced. We acknowledge its role as a supportive tool in the research process.

