# OpenReview forum: "DGSM-SCAM-GAT and MMT-ViT: Multimodal and Graph-Based Malware Detection"
_ICLR.cc/2026/Conference — ICLR 2026 Conference Desk Rejected Submission_

### Official Review · Reviewer_T1rh · 2025-10-20

**Soundness:** 2
**Presentation:** 1
**Contribution:** 2
**Rating:** 2
**Confidence:** 4

**Summary:**

This paper introduces two innovative models, DGSM-SCAM-GAT and MMT-ViT, for malware detection. These models excel in capturing intricate behavioral patterns in API call sequences and fusing features from multiple modalities for effective malware identification.

**Strengths:**

- DGSM-SCAM-GAT and MMT-ViT present two approaches to malware detection, integrating graph-based and multimodal techniques effectively.

- Both models demonstrate accuracy and F1 scores, outperforming existing methods .

- By combining dynamic and static analysis, the models offer an solution for malware detection.

**Weaknesses:**

- This paper lacks a clear and coherent structure, making it challenging for readers to follow the flow of information and understand the key contributions of each model. Clear subsections and a more logical progression would enhance readability.

- The models show a degradation in performance on new datasets, indicating potential issues with generalization. Further validation across diverse datasets with varying characteristics is needed to assess the models' adaptability.

- The high computational cost and training time may limit the practicality and scalability of the models. Detailed information on computational requirements, optimization strategies, and potential trade-offs would provide a clearer understanding of the models' feasibility in real-world settings.

- Although the weighted loss function addresses class imbalance, details on the impact of class weights on model performance, potential biases introduced, and strategies to mitigate these biases are necessary for a comprehensive evaluation.

**Questions:**

NA

---

### Official Review · Reviewer_7HuM · 2025-10-27

**Soundness:** 2
**Presentation:** 3
**Contribution:** 1
**Rating:** 2
**Confidence:** 5

**Summary:**

This paper proposes two models for malware detection problem, and shows the effectiveness with experiments on Kaggle benchmark dataset. The DGSM-SCAM-GAT model combines dynamic gating, contextual aggregation mechanisms, and graph attention networks (GAT)
to improve temporal and structural modeling of API call sequences, and MMT-ViT model uses multi-modal attention mechanisms and the pre-trained ViT to combine features from assembly instruction sequences, binary grayscale images, and binary wavelet sequence features. Both models demonstrate superiority on performance like F1 score to some of the comparing methods.

**Strengths:**

- Overall organization of the paper looks sound and follows the standard format for proposing a new method for malware detection.

- The performance of the proposed models is superior to some of the comparing methods.

**Weaknesses:**

- The novelty of the proposed method is rather weak, and mostly based on combination of previous methods. In this sense, the contribution might be incremental.

- The datasets used for experiments are too limited to show the usefulness in the area of malware detection. Larger and newer datasets are required.

- The comparing methods are outdated. Recent methods for malware detection should be used to compare.

**Questions:**

- What is the main contribution of this work for the representation learning? Simply presenting two methods by combining previous techniques could not be regarded as a novel research.

- What is the contribution of this work to the field of malware detection? The experiment setup is way below the standard of malware detection. Simply reporting on the performance could not be enough for the field of malware detection.

---

### Official Review · Reviewer_HdBU · 2025-10-28

**Soundness:** 2
**Presentation:** 1
**Contribution:** 1
**Rating:** 0
**Confidence:** 4

**Summary:**

This paper proposes two novel architectures: the former (DGSM-SCAM-GAT) for malware detection and the latter (MMT-ViT) for malware classification. The paper claims to overcome state-of-the-art architectures in both tasks.
DGSM-SCAM-GAT integrates Graph Attention Networks, Dynamic Gated Sequence Module, and Contextual Aggregation Module to perform malware detection learning from API call sequences.
MMT-ViT leverages multimodal Transformers and Vision Transformers to fuse instruction sequences, grayscale images, and wavelet sequences to perform malware family classification.
To motivate architectural choices, the paper contains ablation studies in order to justify the presence of each component.
The paper claims to beat the SoA approaches for both tasks with the proposed architectures.

**Strengths:**

+ New hybrid architectures with models and techniques that are not very common in the malware detection and classification domain
+ Trying to propose an approach that consider both problems at once

**Weaknesses:**

The paper does not present any clear and supported objective. The comparison with the state of the art is confused and not very well referenced. Plus, since the improvement is quite negligible, the authors should provide more results on bigger datasets, reporting also the standard deviation of the proposed metrics. Moreover, since a novel architecture is proposed, other properties should be studied: readers do not have a glimpse of how this architecture is resilient to adversarial examples and how costly the training and inference are for such big architectures. For these missing evaluations, the authors should have provided their reasons in a Limitations section.

The cited dataset “big2015” has been used for both detection and classification, for my understanding, but the reference (Ronen, Royi, et al. "Microsoft malware classification challenge." arXiv preprint arXiv:1802.10135 (2018)) reports only malware samples inside it, as it was published as a malware classification challenge. Other works confirm the fact that it can only be used for malware classification tasks (Kalash, Mahmoud, et al. "Malware classification with deep convolutional neural networks." 2018 9th IFIP international conference on new technologies, mobility and security (NTMS). IEEE, 2018).

How was the detection task performed?

The ablation studies of DGSM-SCAM-GAT and MMT-ViT are very minimal and do not support the conclusions, as the small difference in terms of accuracy may be due to experimental variance. Also, authors should comment more on the performance drop that arises from removing components of the architectures.

In general, I would recommend reconsidering the overall structure of the paper:
-	Extend the state of the art and clarify your objectives: why is a new architecture needed to perform malware detection and classification? What are the open problems in these fields?
-	Better describe the dataset used, stating that you renamed it “big2015” dataset in the paper. Moreover, the paper should better describe how the detection task is performed, since the referenced dataset is a malware classification one.
-	Report the standard deviation of each metric, especially when claiming a small improvement in accuracy with respect to the other works.
-	Add statements in a Limitation section on why the adversarial robustness of the DGSM-SCAM-GAT architecture was not tested, for instance, by stating if such attacks exist or not, if they are implemented or not.
-	Improve captions of tables and images for a better understanding.
-	Improve the readability of the paper. There are a lot of missing spaces between words and references (and some of them are broken and not understandable), and the flow of the discussion is very fragmented.

**Questions:**

1) How the detection task has been investigated, since the used dataset does not contain goodware samples?
2) Which are the limitations of this work?

---

### Official Review · Reviewer_BGhd · 2025-10-30

**Soundness:** 2
**Presentation:** 2
**Contribution:** 2
**Rating:** 2
**Confidence:** 3

**Summary:**

This paper introduces two complementary malware detection models – DGSM-SCAM-GT for dynamic API-sequence analysis and MMT-ViT for static multimodal analysis. DGSM-SCAM-GAT combines a dynamic gated sequence module, a contextual aggregation module, and graph attention networks to model API sequences as temporal graphs for binary classification. MMT-ViT fuses assembly instructions, grayscale bytecode images, and wavelet coefficients using a multimodal transformer and ViT backbone for nine-class classification on Big 2015 dataset. Both models report ~99.5-99.6% F1-Score – on separate datasets (a 20k sample dynamic API call sequence dataset and BIG2015), and ablations show modest but consistent performance drops when components are removed.

**Strengths:**

- Strong empirical results and thorough ablations: The reported results show substantial and consistent improvement over baselines such as CNN-LSTM, DenseNet, and Malcse, showing meaningful performance gains in accuracy and F1-score. Ablation studies for both the models are thorough and demonstrate the contribution of each module, validating the design choices.

- Dual Design: The paper correctly identifies the practical divide between dynamic (real-time behavioral) and static (offline feature-based) malware analysis and proposes one model for each domain. This separation, combined with the complementary framing, shows awareness of real-world deployment constraints.

- Practical relevance and conceptual motivation: The dual-model formulation reflects real-world deployment settings where dynamic analysis helps in real-time detection while static multimodal analysis allows deeper offline classification. This is an important and timely framing within malware research.

- Evidence of generalization and transferability: The appendix includes fine-tuning experiments showing strong transfer from the initial training dataset to external dataset (e.g., MalAPI-2019 and Malimg). The improvement from 77.87% to 98.58% after fine-tuning indicates promising adaptability across domains.

**Weaknesses:**

- Limited dataset transparency for DGSM-SCAM-GAT: The dynamic API call sequence dataset is only described as “from Kaggle”, with no dataset name, or preprocessing details.  More insight into the dataset, like provided in the case of Big 2015, would be appreciated to ensure reproducibility and fair evaluation.

- Overly equation-heavy section: The Feature extraction step in 3.2.1. Assembly Sequence Branch is overly equation heavy and could benefit from more intuitive explanation. While technically complete, this reduces readability.

- Lack of information on the implementation of complementary framework: While the paper claims as one of the contributions the implementation of a “complementary approach for dynamic real-time processing and static offline detection” , the two models are trained and evaluated separately, with no integrated pipeline, ensemble, or joint experiment specified. More details on how these components interact – or plans for integration – would strengthen the contribution.

- Limited interpretability and analysis of learned representations: The work focuses on classification performance, but some insight into what structural or semantic information each branch learns could be useful. Visualizing attention maps or modality contributions could provide more interpretability and justify architectural complexity.

**Questions:**

1. Is there a reason why the DGSM-SCAM-GAT dataset details and structure are not disclosed in detail?

2. Were any attempts made to evaluate joint inference – e.g., combining predictions from both models on a sample set?

---

### Note · Program_Chairs · 2026-01-17
**Submission Desk Rejected by Program Chairs**

The following references in this submission do not refer to real documents and/or have major errors in bibliographic information:

 A. Author1, B. Author2, and C. Author3. Wavelet-based feature extraction for malware detection. IEEE Transactions on Dependable and Secure Computing, 18(4):1567-1580, 2021. doi: 10.1109/ TDSC.2020.2983563.